# Cinnamaldehyde Protects against *P. gingivalis* Induced Intestinal Epithelial Barrier Dysfunction in IEC-6 Cells via the PI3K/Akt-Mediated NO/Nrf2 Signaling Pathway

**DOI:** 10.3390/ijms25094734

**Published:** 2024-04-26

**Authors:** Chethan Sampath, Sasanka S. Chukkapalli, Abhinav V. Raju, Leela Subhashini C. Alluri, Dollada Srisai, Pandu R. Gangula

**Affiliations:** 1Department of Diabetes, Metabolism and Endocrinology, Vanderbilt University Medical Center, Nashville, TN 37232, USA; chethan.sampath@vumc.org; 2Department of ODS & Research, School of Dentistry, Meharry Medical College, Nashville, TN 37208, USA; dsrisai@mmc.edu; 3Department of Biomedical Engineering, Texas A&M University, College Station, TX 77843, USA; schukkapalli@tamu.edu; 4College of Osteopathic Medicine, Kansas City University, Kansas City, MO 64106, USA; abhinav.raju@kansascity.edu; 5Department of Periodontics, School of Dentistry, Meharry Medical College, Nashville, TN 37208, USA; lalluri@mmc.edu

**Keywords:** *P. gingivalis*, cinnamaldehyde, intestinal epithelial barrier, oxidative stress, inflammation, nitric oxide, Nrf2

## Abstract

*Porphyromonas gingivalis* (*Pg*), a Gram-negative oral pathogen, promotes and accelerates periodontitis-associated gut disorders. Intestinal epithelial barrier dysfunction is crucial in the pathogenesis of intestinal and systemic diseases. In this study, we sought to elucidate the protective role of cinnamaldehyde (CNM, an activator of Nrf2) against *P. gingivalis* (W83) and *Pg*-derived lipopolysaccharide (*Pg*-LPS) induced intestinal epithelial barrier dysfunction via antioxidative mechanisms in IEC-6 cells. IEC-6 (ATCC, CRL-1592) cells were pretreated with or without CNM (100 µM), in the presence or absence of *P. gingivalis* (strain W83, 10^9^ MOI) or *Pg*-LPS (1, 10, and 100 µg/mL), respectively, between 0–72 h time points by adopting a co-culture method. Intestinal barrier function, cytokine secretion, and intestinal oxidative stress protein markers were analyzed. *P. gingivalis* or *Pg*-LPS significantly (*p* < 0.05) increased reactive oxygen species (ROS) and malondialdehyde (MDA) levels expressing oxidative stress damage. *Pg*-LPS, as well as *Pg* alone, induces inflammatory cytokines via TLR-4 signaling. Furthermore, infection reduced Nrf2 and NAD(P)H quinone dehydrogenase 1 (NQO1). Interestingly, inducible nitric oxide synthase (iNOS) protein expression significantly (*p* < 0.05) increased with *Pg*-LPS or *Pg* infection, with elevated levels of nitric oxide (NO). CNM treatment suppressed both *Pg*- and *Pg*-LPS-induced intestinal oxidative stress damage by reducing ROS, MDA, and NO production. Furthermore, CNM treatment significantly upregulated the expression of tight junction proteins via increasing the phosphorylation levels of PI3K/Akt/Nrf2 suppressing inflammatory cytokines. CNM protected against *Pg* infection-induced intestinal epithelial barrier dysfunction by activating the PI3K/Akt-mediated Nrf2 signaling pathway in IEC-6 cells.

## 1. Introduction

Periodontal disease (PD) is a common inflammatory disease detrimental to digestive and metabolic function. Periodontal pathogens, such as *Porphyromonas gingivalis* (*Pg*), *Fusobacterium nucleatum* (*Fn*), and *Aggregatibacter actinomycetemcomitans*, are the predominant bacterial species of the anaerobic microbiota. These periodontal pathogens are associated with the progression of several systemic diseases such as cardiovascular disease, diabetes, Alzheimer’s disease, and arthritis [1]. The mouth separates internal and external environments and is one of the entry points for bacteria into the gastrointestinal tract, along with saliva, food, and water [2]. Oral pathogenic bacteria are enriched with virulence factors and have adapted to thrive in an inflammatory environment. The salivary abundance of *P. gingivalis* is noted to be significantly higher in patients with periodontal disease (PD) [3]. It was estimated that patients with severe PD swallowed approximately 10^12^~10^13^ *P. gingivalis* bacteria (a major periodontal pathogen) per day [4]. Due to its acid-resistant nature, *P. gingivalis* may pass through the stomach and reach the intestinal tract, destroying intestinal homeostasis [5]. Within the intestine, the intestinal barrier acts as a multifunctional system serving as a microbial, mucous, physical, and immune barrier. The intestinal barrier forms the body’s first line of defense against external pathogens; its breakdown can lead to pathological changes in the gut and other organs or systems [5]. It has been reported that periodontal pathogens may induce multiple changes in the microbiota community, barrier function, and immune system of the gut, which leads to an increased risk of many systemic diseases associated with low-grade inflammation.

The intestinal epithelial barrier depends on specialized structures of intercellular junctions (IJs), including tight junctions (TJs) and adherens junctions (AJs) [6]. TJs, which serve as the interfaces between the apical and basolateral plasma membrane domains, are the primary determinants of paracellular permeability [7]. TJs are composed of various proteins, among which the transmembrane protein occludin (OC-1) and cytoplasmic protein zonula occludens-1 (ZO-1) maintain their structure and intestinal epithelial barrier function [8]. However, it is unknown whether *Pg* can influence these tight junctions and damage gut epithelial barrier function by increasing inflammation and oxidative stress.

Intestinal permeability is regulated by multiple factors including exogenous factors, epithelial apoptosis, cytokines, and immune cells. Oxidative stress mediated by the generation of reactive oxygen species (ROS) is an early-stage trigger of intestinal inflammation [9]. Nrf2 (NF-E2-related factor 2) is a master regulator of cellular responses against environmental stresses. Nrf2 induces the expression of metabolic detoxification and antioxidant enzymes to eliminate ROS. Recent studies have found that the Nrf2-Keap1 pathway participates in many other cellular protective mechanisms, in addition to countering oxidative stress, including the regulation of inflammatory pathways and TJ proteins in the intestinal barrier [10]. Collectively, the multifaceted role of Nrf2-Keap 1 in various types of cells is beneficial in managing several other clinical manifestations [11]. Essential oils have been shown to increase digestive secretions, reduce the number of pathogenic bacteria, and boost the immune system [12]. Cinnamaldehyde (CNM), one of the bioactive essential oils derived from cinnamon, is known for its activation and translocation of Nrf2 [13,14,15]. It has been proven to possess anti-inflammatory, anti-bacterial, and antihypertensive bioactivity [16]. Although a large number of ingested oral bacteria reach GI constantly, the effect of ingested periodontal pathogens such as *P. gingivalis* on intestinal permeability and inflammation is still unknown. Therefore, this study aimed to investigate the molecular mechanism and the underlying protective effects of CNM against *Pg*- and one of its virulence factors, lipopolysaccharide (LPS), induced intestinal epithelial inflammation and barrier dysfunction in IEC-6 cells.

## 2. Results

To evaluate the potential inherent toxicity of the *Pg*/*Pg*-LPS/CNM against IEC-6 cells, the cells were incubated at different concentrations and time intervals (6–72 h), after which cell viability was determined using an MTT assay. The results of cell viability assays presented in Figure 1A show that *Pg*-LPS at all concentrations following 48 h incubation had a negligible effect on the viability of IEC-6 cells. However, cell viability decreased to 80 ± 3% and 65 ± 2% at 10 and 100 µg/mL at 72 h of incubation, respectively, whereas *Pg* alone had no effect until 12 h, but was seen to exert toxicity at 24 h and was more severe at 48 and 72 h (Figure 1A). Overall, cytotoxicity of *Pg* increased with time. Moreover, CNM treatment at the concentrations of 25, 50, and 100 µM showed no significant (*p* < 0.05) effect on the viability of IEC-6 cells compared to the untreated cells at all concentrations and incubation time (Figure 1B).

### 2.1. Protective Effects of Cinnamaldehyde on Pg, Pg-LPS-Induced Cytotoxicity in IEC-6 Cells

To determine the protective effects of the CNM on IEC-6 cells from *Pg*- or *Pg*-LPS- induced cell death, cultures were pre-treated with CNM (25, 50, and 100 µM) for 4 h before co-treatment with *Pg*-LPS at 10 and 100 µg/mL or *Pg* for a 24 to 72 h period. As shown in Figure 1C–E, pretreatment with CNM at all concentrations tested prior to exposure to *Pg* or *Pg*-LPS showed an increase in cell viability. When compared to the other two concentrations (25 and 50 µM), CNM at 100 µM showed an effective response. Thus, these results exhibit that CNM protects periodontal toxicity exerted on IEC-6 cells. Further experiments were carried out for 72 h and *Pg*-LPS at 10 and 100 µg/mL, whereas CNM at 100 µM, based on the results obtained. Overall, the above data shows that *Pg*-LPS at higher concnetartion and with time would significantly reduce cell viability, whereas CNM showed protective role at these concentrations.

### 2.2. Cinnamaldehyde Inhibited Pg or Pg-LPS-Induced Oxidative Stress in IEC-6 Cells

ROS production, MDA content, and NO release from IEC-6 cells were determined to evaluate the potential antioxidant effects of CNM. The data depicted in Figure 2A showed that ROS levels were significantly (*p* < 0.05) increased in IEC-6 cells with *Pg*-LPS or *Pg*. Pretreatment with CNM notably prevented *Pg*-LPS- or *Pg*-induced ROS generation (Figure 2A). Furthermore, the levels of MDA and NO were markedly increased with *Pg*-LPS or *Pg* (Figure 2B,C). When cells were pretreated with CNM, the levels of MDA and NO significantly decreased. 

### 2.3. Cinnamaldehyde Activates Nrf2 Translocation via PI3/Akt Pathway

Next, we investigated whether CNM activates Nrf2. As shown in Figure 3, *Pg*-LPS as well as *Pg* infection induces Keap-1 and suppresses Nrf2 expression, whereas CNM reduces the protein expression of Keap-1, the negative regulator of Nrf2 phase II antioxidant enzyme genes, and increases Nrf2 protein levels. Furthermore, we investigated the downstream Nrf2 protein regulator effects on NAD(P)H quinone oxidoreductase 1 (NQO1). NQO1 is a phase II stress response protein that modulates the production of ROS and can alleviate oxidative stress injury induced by *Pg*-LPS or *Pg* infection. Similarly, activation of Nrf2 by CNM treatment upregulated NQO1 protein expression in IEC-6 cells exposed to *Pg*-LPS or *Pg* infection.

The PI3K/Akt pathway plays a key role in regulating Nrf2-ARE-dependent protection against oxidative stress in IEC-6 cells [17]. Our data demonstrate that *Pg*-LPS or *Pg* infection inhibited phosphorylation of PI3K and Akt, although total PI3k/Akt protein levels showed no significant changes in IEC-6 cells (Figure 3D,E). Pretreatment with CNM stimulated cell proliferation and increased protein levels and nuclear translocation of Nrf2 (Figure 3). This data further confirms the antioxidative properties of CNM at a molecular level.

### 2.4. Cinnamaldehyde Inhibits the Expression of Inducible Nitric Oxide Synthase (iNOS)

IEC-6 cells incubated with *Pg*-LPS or infected with *Pg* resulted in a significant (*p* < 0.05) increase in the expression levels of iNOS (Figure 4A). GTP cyclohydrolase I (GCH-1) is a rate-limiting enzyme responsible for tetrahydrobiopterin (BH_4_) biosynthesis via the de novo pathway [18]. BH_4_ is a co-factor for NO synthesis [18]. Our results show that protein expression of GCH-1 significantly reduced with *Pg*-LPS and *Pg* infection (Figure 4B). Our data further showed that pretreatment with CNM suppressed iNOS and activated GCH-1. *Pg*-LPS at 10 ug/mL did not show good response in iNOS/ROS signaling. *Pg*-LPS at 100 ug/mL as well as *Pg* alone were more robust.

### 2.5. Cinnamaldehyde Reduces the Expression Levels of Proinflammatory Cytokines

Pro-inflammatory cytokines act as agents to invade gut-barrier integrity [19]. Our data showed gene transcripts of tumor necrosis factor-alpha (TNF-α), interleukin (IL)-1beta (β), IL-6, and toll-like receptor-4 (TLR-4) significantly increased in response to *Pg*-LPS or invasion of *Pg*, respectively (Figure 5A–D). We next evaluated the protein expression for these markers. The results correlated with higher expression of TLR-4, Nfkb, and IL-1β by *Pg*-LPS- or *Pg*-infected epithelial cells (Figure 5E–G). Whereas pre-incubation with CNM significantly reduced the expression levels of pro-inflammatory cytokines.

### 2.6. Cinnamaldehyde Increased Occludin and ZO-1 Protein Expression and Intestinal Epithelial Barrier Function in IEC-6 Cells

Epithelial TJs increase intestinal epithelial barrier function, so the expression of two representative TJ proteins, ZO-1, and occluding (OC-1), were analyzed by western blot. ZO-1 and occludin protein expression was significantly reduced after treatment with *Pg*-LPS or *Pg* infection (Figure 6A,B). ZO-1 and OC-1 protein expression were significantly upregulated after treatment with CNM (Figure 6A,B).

Additionally, we investigated the effects of *Pg*-LPS or *Pg* on intestinal TJ permeability by measuring FITC-dextran (FD4) flux in IEC-6 cells. FD4 flux was significantly increased in the cells treated with *Pg*-LPS or *Pg* (Figure 6C). However, CNM attenuated *Pg*-LPS- or *Pg*-induced intestinal TJ permeability.

### 2.7. Cinnamaldehyde Inhibits Bax/Caspase-3 Signaling Pathway in IEC-6 Cells Infected with Pg-LPS/P. gingivalis

The Bcl-2 and caspase families serve an important role in the regulation of cellular apoptosis [20]. In order to characterize the mechanism by which the infection with *Pg*-LPS *or Pg* induces IEC-6 cellular apoptosis, the protein expression levels of Bcl-2 and Bax were detected by western blotting. Compared to the control, the relative protein expression levels of caspase-3 and Bax were significantly increased, and the relative expression of Bcl-2 protein decreased significantly in IEC-6 cells following infection with *Pg*-LPS and *Pg* respectively (Figure 7). These results suggest that infection with *Pg*-LPS and *Pg* may have promoted cellular apoptosis via the Bcl-2/Bax signaling pathway.

## 3. Discussion

Periodontal pathogens such as *P. gingivalis* induce multiple changes in the microbiota community, barrier function, and immune system of the gut, which lead to an increased risk of many systemic diseases associated with low-grade inflammation and eventually gastrointestinal (GI) dysbiosis [21,22]. Interactions between invasive periodontal pathogens and host cells have provided new insights into the pathogenesis of periodontal disease. Modulation of the mucosal epithelial barrier by pathogenic bacteria appears to be a critical step in the initiation and progression of periodontal disease. Upregulation of genes in antioxidative signaling pathways, such as the Nrf2-antioxidant response element (ARE)-NQO1, reduces systemic inflammatory burden and protects against GI tract disorders. The permeability of human intestine is more than *in vitro* models. Therefore, the experiments were conducted using rat IEC 6 cells which are similar in permeability to human intestine. In the present study, we demonstrated that CNM showed beneficial effects on intestinal epithelial barrier function via the PI3K/Akt-mediated Nrf2 antioxidant signaling pathway in IEC-6 cells.

The GI tract is a key source of ROS, which is involved in many GI diseases [23]. It has been reported that *P. gingivalis* induces the rapid production of ROS in gingival epithelial cells. Excessive production of ROS due to oxidative stress alters cellular proteins and their functions, leading to cellular dysfunction and disruption [24]. ROS can break the lipid membrane and increase membrane fluidity and permeability causing intestinal epithelial barrier dysfunction [24,25]. It is evident that periodontal pathogens such as *Pg* induce the rapid production of ROS in brain endothelial cells [26]. ROS triggers peroxidation of membrane lipids, resulting in the formation of malondialdehyde (MDA) and acetaldehyde (AA) [27]. ROS-related tissue destruction could be observed as the final product of lipid peroxidation, such as MDA. In an in vivo study, animals inoculated with *Pg* significantly increased levels of MDA in the myocardium [28]. A previous study demonstrates that ROS oxidizes BH_4_ to dihydrobiopterin (BH_2_), resulting in eNOS uncoupling and O_2_ as a byproduct rather than NO [29]. Elevated levels of NO production cause epithelial cell cytotoxicity and pulmonary epithelial cell damage [29]. Another study reported that excessive NO may react with superoxide anion radicals, giving rise to strong oxidizing agents, such as peroxynitrite, which consequently destroy functional tissues [30]. From our studies, we have demonstrated that increased oxidative stress and reduced NO levels in gastric tissues of diabetic mice led to gastric motility disorders [14]. In this study, LPS derived from *P. gingivalis* or *P. gingivalis* significantly increased the levels of ROS, MDA, and NO in IEC-6 cells. Pretreatment with CNM remarkably decreased ROS production, MDA, and NO levels. These findings indicated that CNM increased the antioxidant capacity and prevented the damage of oxidative stress induced by *Pg*-LPS or *Pg* infection in IEC-6 cells.

Oxidative stress is induced by an imbalance between production of ROS and the antioxidant defense system. Nrf2-regulated redox balance is perturbed in several inflammatory diseases [31]. The protective effect of Nrf2 in maintaining the barrier has been proved in various experimental models, including *P. gingivalis* infection, dextran sodium sulfate-induced colitis, intestinal ischemia-reperfusion, aspirin/NSAID-induced vascular damage, intestinal burn, severe sepsis, and traumatic brain injury-induced intestinal mucosa damage and epithelial barrier dysfunction [10]. The results from the present study demonstrated that Nrf2, Keap1, and NQO1 expression levels were significantly reduced, and the increased rate of apoptosis causing epithelial barrier damage in infected cells exposed to *Pg* or its virulence factor LPS compared to that of the non-infected cells. These findings indicate that the oxidative stress caused by periodontal bacterial infection leads to cell injury and apoptosis suggesting that the Nrf2-Keap1-ARE-NQO1 signaling pathway plays an essential role in periodontal disease.

The PI3K/AKT pathway regulates a number of biological processes including cell growth, differentiation, apoptosis, and neoplastic transformation [32]. Furthermore, the PI3K/Akt signaling pathway is affected by *Pg* and its virulence factor LPS, fimbriae, and gingipains [32]. The results of the present study suggested that CNM remarkably increased the phosphorylation level of Akt protein, which contributes to CNM-induced Nrf2/NQO-1 protein upregulation. This finding indicated that the PI3K/Akt signaling cascade is associated with the protective effects of CNM on intestinal epithelial barrier function via the Nrf2/NQO1 antioxidant pathway in IEC-6 cells.

NOS catalyzes the synthesis of NO in vivo; under normal physiological states, NO synthesis mediated by endothelial or gastric neuronal NOS serves a major role in the regulation of vascular or gastric motility [33]. High levels of NO secretion and the expression of iNOS increase correspondingly following severe bacterial infection [34]. Kidney epithelial cells also have been shown to express iNOS, in animals and in humans when stimulated with LPS [35]. We reported earlier that *Pg* infection in primary human aortic endothelial cells showed a significant (*p* < 0.05) increase in iNOS mRNA levels [29]. Our results clearly show higher levels of NO secretion that correlate with iNOS expression.

Chen et al. demonstrated that *Pg* infection induced an inflammatory reaction in lung epithelial cells, promoting the accumulation of NO, which results in damage to lung epithelial cells and the induction of apoptosis [36]. It was hypothesized that *P. gingivalis* may induce the expression of cell adhesion molecules, TLRs, chemokines, and a large number of inflammatory cytokines. LPSs are the major inflammatory mediators for Gram-negative bacteria. *Pg*-LPS is able to elicit cell inflammatory responses via interaction with TLRs [37]. The lipid A component is responsible for LPS binding to myoloid differentiation factor 2, the resulting complex then binds to TLR4 and triggers a signaling cascade, leading to the production and secretion of pro-inflammatory cytokines [37]. TLR4 is combined with MyD88, phosphorylated IRAK-4 activates the TRAF6-TAK1-NF-κB/MAPK signaling axis which promotes the release of inflammatory factors, such as IL-1β, IL-6, and most importantly TNF-α [38]. The results of the present study revealed that cytotoxic products, such as the LPS of *P. gingivalis*, induce host cells through TLR4, NF-κB activation, and to express TNF-α, IL-1β, and IL-6, thus, synthesis of NOS is induced to increase the production of NO, which leads to epithelial cell damage. Our results also demonstrate that CNM attenuated the intestinal inflammatory responses by inhibiting the NF-κB signaling pathway in IEC-6 cells infected with *Pg*. These results are in line with the previous study which reported that cinnamaldehyde suppressed the production of TNF-α and IL-6 in ulcerative colitis, as well as high-fat diet-induced diabetic mouse models and in vitro studies [14,16,39].

Tomofuji et al. reported that oxidative DNA damage of the liver was found in rats with lipopolysaccharide/protease-induced periodontitis, with increasing serum levels of hydrogen peroxide [40]. ROS play an important role in ischemic tissue injury and the pathogenesis of a number of intestinal disorders, such as inflammatory bowel disease (IBD) and necrotizing enterocolitis (NEC) [41]. Previous reports suggest that increased ROS is associated with impaired epithelial barrier function [41]. Excessive oxidative stress will result in intestinal inflammation and apoptosis of intestinal mucous epithelium, which further damages the intestinal mucosa barrier [10]. TJs are the primary determinants of paracellular permeability. Previous studies have shown that the zonula occludins (ZO), claudin family proteins, and occludin are essential components of TJs in the epithelial barrier. Occludin and ZO-1 proteins have been implicated in maintaining TJ structure and intestinal epithelial barrier function. A significant body of evidence indicates that TJs are associated with various intracellular signaling molecules and are controlled by many signaling pathways [42]. A recent study showed that CNM treatment effectively enhances intestinal barrier integrity, ameliorates inflammatory responses, and remodels gut microbiome in early-weaned rats [16]. Therefore, the present study showed that CNM significantly increased the protein levels of occludin and ZO-1 in *Pg*-infected IEC-6 cells.

Apoptosis or programmed cell death is triggered by two distinct signaling pathways; the intrinsic or stress-activated and the extrinsic or receptor-activated apoptotic pathway [43]. In vitro studies show that *P. gingivalis* can modulate apoptosis in various cell types: such as fibroblasts, endothelial cells, and lymphocytes, and apoptosis has been proposed as a mechanism to explain the extensive tissue destruction in chronic periodontitis lesions [44]. The Bcl2 family of proteins governs the intrinsic pathway of apoptosis. Apoptosis is inhibited when the proapoptotic members of the Bcl2 family insert their BH3 domain into the hydrophobic groove, the former by BH1-4 domains, of the antiapoptotic member Bcl2 [45]. In the present study it was observed that compared with the control, the relative protein expression levels of caspase-3 and Bax increased significantly, and that the relative expression levels of Bcl-2 protein decreased significantly, following infection of the cells with *Pg* and *Pg*-LPS. The results from this study revealed that pretreatment with CNM has inhibited cellular apoptosis by modulating the Bcl-2/Bax/caspase-3 signaling pathway.

## 4. Materials and Methods

### 4.1. Pre-Culture of Pg

*Pg* was cultured as described previously [29]. Briefly, *Pg* (strain W83) was inoculated into 3 mL Tryptic Soy Agar (TSA) broth supplemented with hemin and menadione, which allowed bacteria to grow until the mid-log phase (OD600 of ≈0.8). All reagents were purchased from ATCC (Manassas, VA, USA) or Fisher Scientific (Hampton, NH, USA).

### 4.2. IEC-6 Cell Culture

The rat small intestinal epithelial (IEC-6) cells were purchased from the ATCC (CRL-1592, Manassas, VA, USA) and maintained in DMEM (Gibco, New York, NY, USA) with 10% fetal bovine serum (FBS, Gibco, NY, USA), 100 U/mL penicillin, 100 mg/mL streptomycin, and 0.01 mg/mL insulin in a humidified (37 °C and 5% CO_2_) incubator.

### 4.3. Human Oxygen-Bacteria Anaerobic (HoxBan) Coculture and Treatment

The Human oxygen-Bacteria anaerobic (HoxBan) coculture method was performed as described previously with minor modifications [46]. Three milliliters of the overnight *Pg* (W83, 10^9^ colony-forming units) pre-culture were used to inoculate freshly autoclaved and cooled-down (≈40 °C) TSA containing 1% agar. Aliquots of 150 mL of this inoculum were transferred into a sterile cell culture dish (145 mm) and allowed to solidify for 30 min. The *Pg*-inoculated culture dish was maintained in an anaerobic chamber. IEC-6 cultured on glass plates (1 × 10^6^ cells) were placed upside down on top of the agar and overlaid with 50 mL of pre-warmed (37 °C) DMEM medium (without antibiotics). For respective controls (uninfected), IEC-6 cultured on glass slides were placed upside down on top of the agar without *Pg*. Following this process, the coculture dishes were placed in a humidified incubator at 37 °C and 5% CO_2_ and incubated at different time points (0 to 72 h). The multiplicity of infection (MOI) was calculated based on the number of cells per well when seeded. Three separate experiments were performed, each performed in duplicate for each condition.

### 4.4. Cell Viability Assay

*Pg*/*Pg*-LPS/CNM-induced cell toxicity was assessed using an MTT colorimetric assay [47]. For this assay, IEC-6 cells were seeded in 96-well plates (5 × 10^4^ cells/well) for LPS (derived from *Pg*) and CNM experiments. For *Pg*, the HoxBan coculture method described above was used. CNM (25–100 µM) was dissolved using cell growth media, whereas LPS derived from *P. gingivalis* (1, 10, and 100 µg/mL) was dissolved in water, and subsequent dilutions were made in growth media. Cells at 70–80% confluence were placed in DMEM media free of FBS. The cells in the media were treated with either *Pg*-LPS or CNM at different concentrations and incubated for 72 h. After exposure to different conditions for 72 h, the media was completely removed, and the cells were incubated with MTT at 0.5 mg/mL (final conc.) for 3 h. The insoluble purple formazan product formation was read at 570 nm using a microplate reader (BioTek, Winooski, VT, USA). The results were expressed as the percentage of control treatment representing 100% viability.

### 4.5. Cinnamaldehyde Treatment

IEC-6 cells (5 × 10^4^ cells/well) were plated in 96-well plates and allowed to attach for 48 h at 37 °C and 5% CO_2_. CNM was prepared using cell growth media. At 70–80% confluence, cells were pre-incubated with CNM (100 μM), which was the concentration selected based on the results from the above experiments, for 4 h, then co-treated with either *Pg*-LPS (10 and 100 µg/mL) or *Pg* and incubated at 37 °C for 24–72 h. The cells were incubated with MTT at 0.5 mg/mL (final conc.) for 3 h. After the incubation period, the media was aspirated, and the absorbance was read at 570 nm using a microplate reader (BioTek, Winooski, VT, USA). The experiment was repeated independently to confirm the results.

### 4.6. Intracellular ROS Determination

Intracellular reactive oxygen species (ROS) levels were detected by a fluorescence probe, 2′,7′-dichlorodihydrofluorescein diacetate (DCFH2-DA; Invitrogen, Carlsbad, CA, USA). Briefly, IEC-6 cells cultured in 48-well plates for 24 h before the experiment, were pretreated with CNM (100 µM) for 4 h and then cocultured with *Pg* or co-treated with *Pg*-LPS (10 and 100 µg/mL) at 37 °C for 6 h. The cells were washed three times with PBS and then incubated with 20 µM DCFH2-DA for 15 min. Intracellular fluorescence intensity was calculated by a fluorescence microplate reader (BioTek, Winooski, VT, USA) at an excitation wavelength of 495 nm and an emission wavelength of 525 nm.

### 4.7. Nitric Oxide and Malondialdehyde Determination

In brief, at the end of the co-culture experiment for each time point as described in the above sections, the cell culture supernatants were collected for nitric oxide (NO) release using a NO assay kit (Biovision, Milpitas, CA, USA) according to the manufacturer’s protocol. Malondialdehyde (MDA) content was also detected from the cell culture supernatants by using a commercial kit (MyBiosource, San Diego, CA, USA), according to the manufacturer’s instructions.

### 4.8. Intestinal Epithelial Barrier Function Determination

The intestinal epithelial barrier function was determined by FD4 flux, as previously described [29]. To investigate intestinal epithelial paracellular permeability, fluorescent-conjugated dye (FD4; 1 mg/mL) was applied to the apical chamber, and the medium was collected from the basolateral chamber after incubation for 2 h at 37 °C in sterile conditions. The fluorescence intensity of FITC was quantified by using a fluorescence plate reader (BioTek, Winooski, VT, USA) with excitation at 492 nm and emission at 520 nm.

### 4.9. RNA Isolation and Quantitative Real-Time Polymerase Chain Reaction Analysis

At the end of the co-culture experiment, IEC-6 cells adherent to coverslips were removed from co-culture tubes, and the total RNA of cells was extracted with TRIzol. The quality of RNA was determined by NanoDrop. RNA was reverse transcribed to cDNA using iScript cDNA Synthesis Kit. A quantitative real-time-polymerase chain reaction assay was performed using SYBR Green. The sequences of primers for target genes are listed in Table 1. Relative expression levels of target genes were normalized to β-actin, and threshold cycle (Ct) numbers were calculated using the 2^−ΔΔCT^ method). All studies were performed in the Meharry Medical College Molecular Biology Core Laboratory.

### 4.10. Western Blot Analysis

IEC-6 cells were lysed in RIPA buffer with protease inhibitors on ice to extract total protein. The protein concentrations were measured by using a BCA protein assay kit (Pierce, Rockford, IL, USA). Equal amounts of protein (50 µg) were separated by SDS-PAGE and then transferred onto a nitrocellulose membrane. After the membranes were blocked with blocking buffer for 1 h, they were incubated with the following primary antibodies, as listed in Table 2. The membranes were then washed and incubated with secondary antibody at a dilution of 1:10,000 for 1 h, min at room temperature. Then, the membranes were visualized by using ImageQuant LAS 500 (GE Health Sciences, Pittsburgh, PA, USA).

### 4.11. Statistical Analysis

The data are represented as the means ± SD. Statistical analyses were performed by using GraphPad Prism 5 software (GraphPad, San Diego, CA, USA). One-way or two-way repeated measures (RM) ANOVA were performed where appropriate, and Tukey or Bonferroni’s post-tests were used to evaluate significant differences. In all cases, the results were considered significant at *p* < 0.05.

## 5. Conclusions

In summary, the present study demonstrated that CNM protected IEC-6 cells against *Pg*-induced oxidative damage and increased intestinal epithelial barrier permeability. Additionally, CNM alleviated intestinal epithelial barrier dysfunction by activating the PI3K/Akt-mediated Nrf2/NQO-1 antioxidant signaling pathway and suppressing inflammatory and apoptotic markers. These results provide new insights into the molecular mechanisms underlying the CNM-mediated protective effects of intestinal epithelial barrier function and have important clinical applications in intestinal barrier dysfunction and periodontal diseases. Finally, more studies are warranted to determine the efficacy of cinnamaldehyde in clinical research.

## Figures and Tables

**Figure 1 ijms-25-04734-f001:**
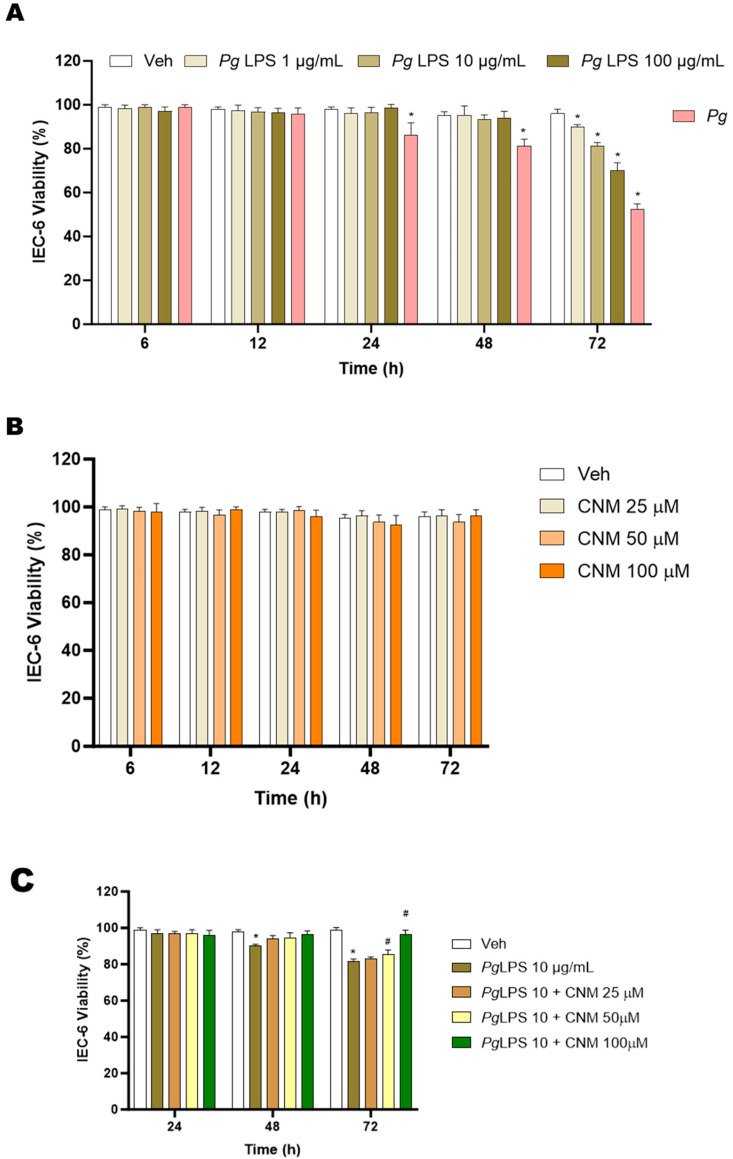
Screening of cytotoxic effects of *Pg*-LPS, *Pg*, and cinnamaldehyde (CNM) on IEC-6 cells. (**A**) IEC-6 cells were treated with 0, 1, 10, 100 µg/mL of *Pg*-LPS, *Pg* alone and (**B**) 25, 50, 100 µM of CNM for 6, 12, 24, 48, and 72 h. The cell growth inhibitory rate was measured using an MTT assay. The values are expressed as a percentage of the control (uninfected with *Pg*-LPS or *Pg*) for each time point, respectively (0 to 72 h). (**C**–**E**) protective effects of CNM (100 µM) against *Pg*-LPS (10 and 100 µg/mL) and *Pg*-induced damage on IEC-6 cell viability. The results from three independent experiments are represented in the form of means ± SD (n = 3, three independent experiments and, in each experiment, we had triplicates values for all the results). * *p* < 0.05 compared with control (non-infected) cells. # *p* < 0.05 compared with control (infected) cells. *Pg*—*P. Gingivalis*; LPS-lipopolysaccharide.

**Figure 2 ijms-25-04734-f002:**
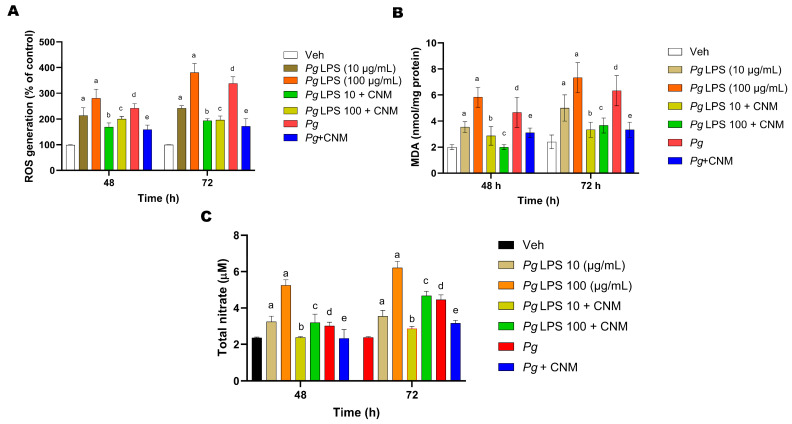
Cinnamaldehyde (CNM) inhibits *Pg*-LPS * and Pg* induced intracellular ROS, MDA, and NO changes in IEC-6 cells. (**A**) IEC-6 cells were pretreated with CNM (l00 µM) for 4 h prior to exposure to *Pg*-LPS or *Pg* at 48 and 72 h. The cells were incubated with DCFH2-DA for 15 min. ROS generation was detected by fluorescence values of ROS were measured by using a fluorescence microplate reader. The fluorescence intensity read at 48 and 72 h for the vehicle was 46.27 ± 12.34 and 57.24 ± 28.61, respectively. (**B**) The cells were pretreated with CNM (100 µM) for 4 h and then exposed to *Pg*-LPS or *Pg* at 48 and 72 h. The MDA levels in the supernatant were determined by using commercial kits. (**C**) The NO concentration in the supernatant was determined by the Griess reaction. Data are presented as the means ± SEM of three independent experiments. a, d: *p* < 0.05 compared to the control group; b: *p* < 0.05 compared to *Pg*-LPS 10 µg/mL, c: *p* < 0.05 compared to *Pg*-LPS 100 µg/mL, e: *p* < 0.01 compared to the *Pg* group.

**Figure 3 ijms-25-04734-f003:**
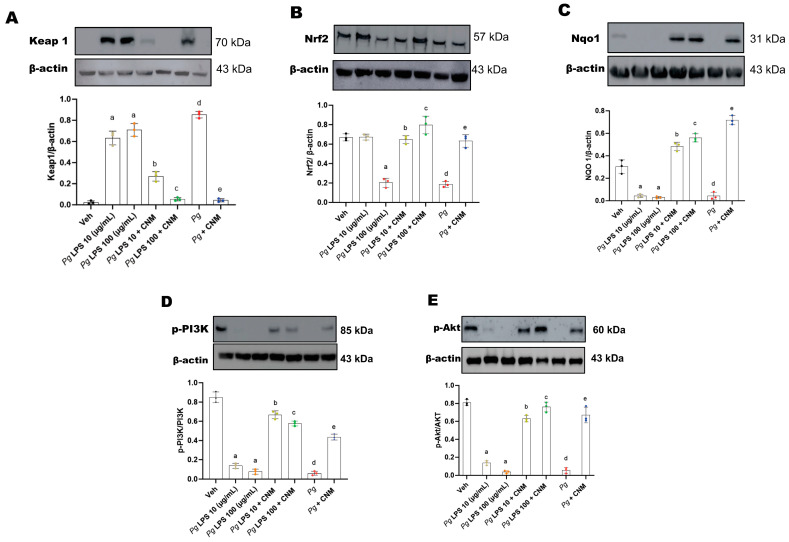
Cinnamaldehyde (CNM) treatment suppresses Keap-1, activates Nrf2, NQO1, and PI3K/Akt phosphorylation protein expression in *Pg*-LPS and *Pg* induced IEC-6 cells. The IEC-6 cell was pre-incubated with CNM (100 µM) for 4 h and exposed to *Pg*-LPS or *Pg* for 72 h. (**A**) Keap-1, (**B**) Nrf2, (**C**) NQO1, (**D**) p-PI3K, and (**E**) p-AKT protein expressions in IEC-6 cells at 72 h. β-Actin was used as a loading control. Bar graphs showed a ratio of target gene or protein with β-actin. Data were analyzed using one-way ANOVA by using GraphPad prism software. Values are mean ± SD (n = 3). a, d: *p* < 0.05 compared to the control group; b: *p* < 0.05 compared to *Pg*-LPS 10 µg/mL, c: *p* < 0.05 compared to *Pg*-LPS 100 µg/mL, e: *p* < 0.01 compared to the *Pg* group.

**Figure 4 ijms-25-04734-f004:**
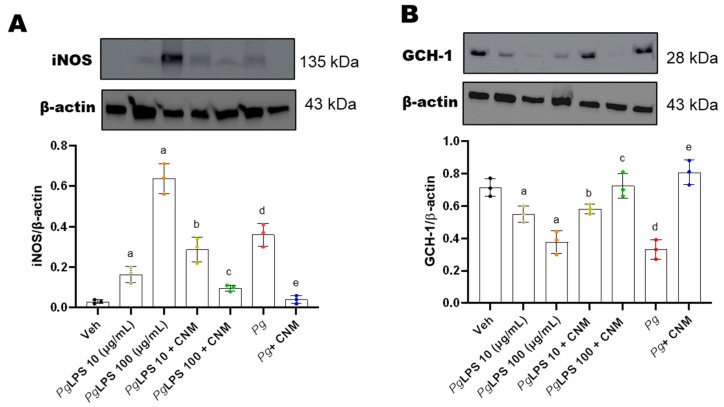
Cinnamaldehyde (CNM) suppresses iNOS and activates GCH-1 protein expression in *Pg*-LPS and *Pg* induced IEC-6 cells. IEC-6 cell was pre-incubated with CNM (100 µM) for 4 h and exposed to *Pg*-LPS or *Pg* for 72 h. (**A**) iNOS and (**B**) GCH-1 protein expression in IEC-6 cells. Stripped blots were re-probed with β-actin. Bar graphs showed a ratio of target protein with β-actin. Data were analyzed using one-way ANOVA by using GraphPad Prism software. Bars represent mean values, with error bars representing SD. Data are for three independent experiments (n = 3). a, d: *p* < 0.05 compared to the control group; b: *p* < 0.05 compared to *Pg*-LPS 10 µg/mL, c: *p* < 0.05 compared to *Pg*-LPS 100 µg/mL, e: *p* < 0.01 compared to the *Pg* group.

**Figure 5 ijms-25-04734-f005:**
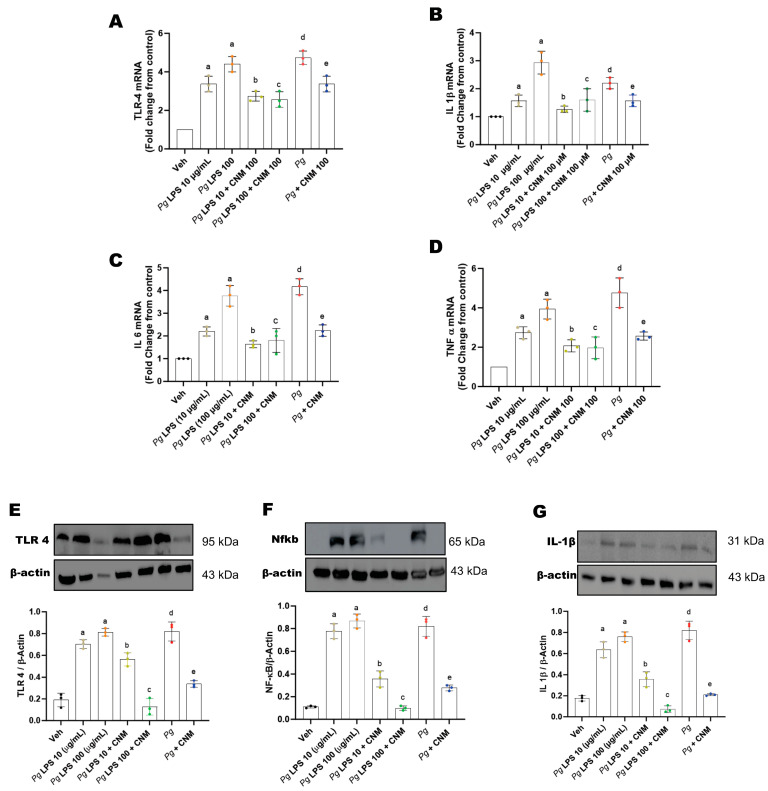
Cinnamaldehyde (CNM) reduces cytokine expression in *Pg*-LPS and *Pg* induced IEC-6 cells. The IEC-6 cell was pre-incubated with CNM (100 µM) for 4 h and exposed either with *Pg*-LPS or *Pg* for 72 h. The cell lysates were subjected to RT-qPCR, as well as western blot analysis. (**A**) TLR-4, (**B**) IL-1β, (**C**) IL 6, and (**D**) TNF α transcript levels. Representative immunoblot and densitometric analysis data for (**E**) TLR-4, (**F**) Nfκb, and (**G**) IL-1β. β-Actin was used as a loading control. Stripped blots were re-probed with β-actin. Bar graphs showed a ratio of target protein with β-actin. Data were analyzed using one-way ANOVA by using GraphPad Prism software. a, d: *p* < 0.05 compared to the control group; b: *p* < 0.05 compared to *Pg*-LPS 10 µg/mL, c: *p* < 0.05 compared to *Pg*-LPS 100 µg/mL, e: *p* < 0.01 compared to the *Pg* group.

**Figure 6 ijms-25-04734-f006:**
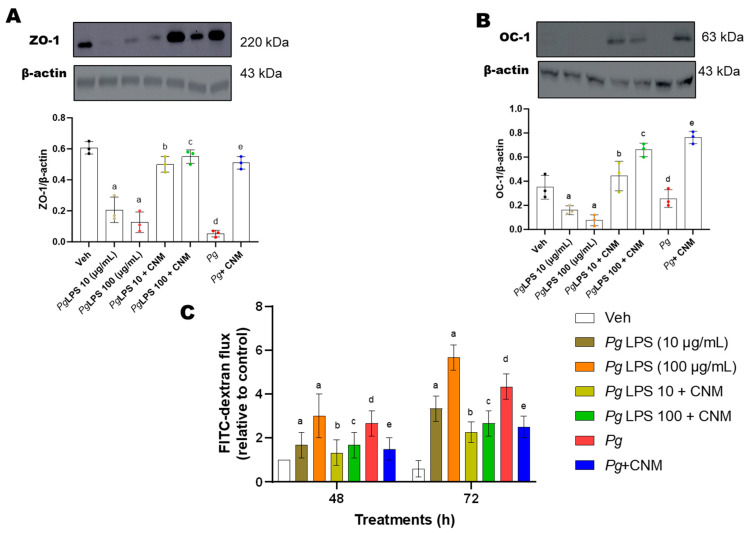
Effects of cinnamaldehyde (CNM) on ZO-1 and OC-1 protein expression and intestinal TJ permeability changes in *Pg*-LPS and *Pg* induced IEC-6 cells. IEC-6 cell was pre-incubated with CNM (100 µM) for 4 h and exposed to either *Pg*-LPS or *Pg* for 72 h. The cell lysates were subjected to RT-qPCR, as well as western blot analysis with an (**A**) anti-ZO-1 and (**B**) anti-OC-1, antibody. (**C**) The permeability of FD4 across the cell monolayer was measured after exposure to *Pg*-LPS and *Pg,* which were pre-treated with cinnamaldehyde (CNM). a, d: *p* < 0.05 compared to the control group; b: *p* < 0.05 compared to *Pg*-LPS 10 µg/mL, c: *p* < 0.05 compared to *Pg*-LPS 100 µg/mL, e: *p* < 0.01 compared to the *Pg* group.

**Figure 7 ijms-25-04734-f007:**
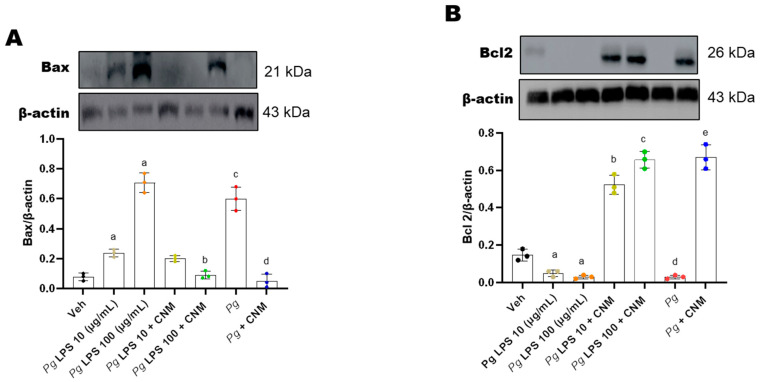
Cinnamaldehyde (CNM) prevents *Pg*-LPS and *Pg* induced apoptosis in IEC-6 cells. IEC-6 cell was pre-incubated with CNM (100 µM) for 4 h and exposed to either *Pg*-LPS or *Pg* for 72 h. Representative immunoblots and densitometric analysis data for (**A**) Bax, and (**B**) Bcl2 in IEC-6 cells. Blots showing same β-actin were stripped and re-probed. Data were normalized with band intensities for β-actin. Bar graphs depict ratios of target proteins to β-actin. Data are for four independent experiments. a, d: *p* < 0.05 compared to the control group; b: *p* < 0.05 compared to *Pg*-LPS 10 µg/mL, c: *p* < 0.05 compared to *Pg*-LPS 100 µg/mL, e: *p* < 0.01 compared to the *Pg* group.

**Table 1 ijms-25-04734-t001:** Primers used for quantitative real-time PCR.

Gene	Forward	Reverse
*IL-1β*	5′-CTTTGAAGCTGATGGCCCTAAA-3′	5′-AGTGGTGGTCGGAGATTCGT-3′
*IL-6*	5′-TCAATGAGGAGACTTGCCTG-3′	5′-GATGAGTTGTCATGTCCTGC-3′
*TLR-4*	5′-GATCTGTCTCATAATGGCTTG-3′	5′-GACAGATTCCGAATGCTTGTG-3′
*β-actin*	5′-ATCCTCACCCTGAAGTACCC-3′	5′-TAGAAGGTGTGGTGCCAGAT-3′
*TNF α*	5′- CACGCTCTTCTGTCTACTGA-3′	5′-ATCTGAGTGTGAGGGTCTGG-3′

**Table 2 ijms-25-04734-t002:** List of antibodies used in this study.

Sl. No.	Antibody	Source
1	Rabbit anti-occludin	#71–1500, Invitrogen
2	Rabbit anti-ZO-1	#61–7300, Invitrogen
3	Rabbit anti-p-Akt	#9271, Cell Signaling Technology (Danvers, MA, USA)
4	Mouse anti-Tlr4	sc-293072, Santa Cruz Biotechnologies (Dallas, TX, USA)
5	Rabbit anti-β-actin	#4970, Cell Signaling Technology
6	Mouse anti-Kelch-like ECH-associated protein 1 (Keap-1)	sc-365626, Santa Cruz Biotechnologies
7	Mouse anti-Nrf2	sc-365949, Santa Cruz
8	Mouse anti-GTP cyclohydrolase 1 (GCH-1)	sc-271482, Santa Cruz
9	Mouse anti-NQO-1	#sc-376023, Santa Cruz
10	Rabbit anti-Bax	#2772, Cell Signaling
11	Rabbit anti-Bcl2	#2876, Cell Signaling
12	Rabbit anti-Nfkb (p65)	#ab16502, Abcam (Cambridge, UK)
13	Mouse anti-IL-1β	#sc-12742, Santa Cruz Biotechnologies
14	Rabbit anti-iNOS	#ab15323, Abcam
15	Rabbit anti-PI3K	#4228, Cell Signaling

## Data Availability

The data generated as a part of the study has been submitted in this article.

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
