# Peer review of "Cinnamaldehyde Protects against P. gingivalis Induced Intestinal Epithelial Barrier Dysfunction in IEC-6 Cells via the PI3K/Akt-Mediated NO/Nrf2 Signaling Pathway"

_ijms, 2024, doi:10.3390/ijms25094734_

Round 1

Reviewer 1 Report

Comments and Suggestions for Authors

This manuscript investigated in the oral–gut microbiota axis where periodontal disease and intestinal and systemic diseases may be intersecting through the impairment of intestinal barrier integrity. This is a topical and very interesting issue. 

Therefore, for this manuscript I suggest some improvements in the text and more careful attention in displaying the results.

Introduction: in line 61 Authors stated that periodontal pathogens are associated with the progression of Alzheimer’s disease and in line 65 that these pathogens are higher in patients with PD relative to other systemic conditions. Please, first Authors have to specify that PD is Parkinson’s disease, then to better clarify if these periodontal pathogens affect the two neurodegenerative diseases equally or are prevalent on one of the two. 

Moreover, if possible, state whether the route of propagation of toxins of the periodontal pathogens to the CNS only through the intestine and then the bloodstream or can also involve the vagus or the trigeminal nerves. In this case, if a direct action of cynnamaldehyde on oral microbiota toxin effects could be also conceivable in the Conclusions.

Results: the first section of the results is confused (lines 97-113). Indeed, on the text results are described on separated Fig. 1a and Fig.1b, but Figure1 consists only of Fig 1A and Fig.1b is missing (please make also the letters uniform if uppercase or lowercase). Furthermore, in Figure 1 the histograms are too crowded and make it difficult to understand the data.

In Figure 2, Authors should report the fluorescence intensity values read with vehicle, Pg-LPS and Pg in the caption of the figure.

In section 2.3, lines 152-153, it is unclear what specific data of the results demonstrate that “pretreatment with CNM stimulated cell proliferation and increased protein levels and nuclear translocation of Nrf2”.

Discussion: please, reference the statement of the line 243 “The GI tract is a key source of ROS, which are involved in many GI diseases.”, and the statement of the lines 280-281 “NOS catalyzes the synthesis of NO in vivo; under normal physiological states, NO synthesis mediated 280 by endothelial or gastric neuronal NOS serves a major role in the regulation of vascular or gastric motility.”.

Specify the meaning of the acronyms IBD (line 305) and NEC (line 306).

Methods: line 330: Pg is of human or rat origin? I mean, if Pg is of human origin, is its effect on rat IEC-6 cells reliable? Authors should better specify why in their experimental design a human periodontal pathogen is tested on rat intestinal cells.

Please, specify the meaning of the acronyms TSA (line 330), MOI (line 347) and FD4 (line 382) for the outsider readers.

Section 4.4:  MTT solution was effectively incubated for 3 h at 5 mg/ml or this one was diluted 1:10?

Conclusions: can it be concluded that the tested concentrations of cinnamaldehyde on IEC-6 cells are actually achievable in vivo in the intestine and, therefore, having the same effects also in the human intestine?

Author Response

Reviewer #1

Comments and Suggestions for Authors

This manuscript investigated in the oral–gut microbiota axis where periodontal disease and intestinal and systemic diseases may be intersecting through the impairment of intestinal barrier integrity. This is a topical and very interesting issue. 

Therefore, for this manuscript I suggest some improvements in the text and more careful attention in displaying the results.

Response: We are thankful for the insightful review and suggestions provided by this reviewer. We have carefully revised the manuscript and the figures as suggested.

Comments:

  1. Introduction: in line 61 Authors stated that periodontal pathogens are associated with the progression of Alzheimer’s disease and in line 65 that these pathogens are higher in patients with PD relative to other systemic conditions. Please, first Authors have to specify that PD is Parkinson’s disease, then to better clarify if these periodontal pathogens affect the two neurodegenerative diseases equally or are prevalent on one of the two. 

Response: Thank you for bringing this to our notice. PD was abbreviated earlier in the introduction, so the authors used PD which refers to periodontal disease. This was revised in the manuscript for better clarity. Revised track-in-changes manuscript line 60-61 has been revised to “The salivary abundance of P. gingivalis is noted to be significantly higher in patients with periodontal disease (PD)”.

1a. Moreover, if possible, state whether the route of propagation of toxins of the periodontal pathogens to the CNS only through the intestine and then the bloodstream or can also involve the vagus or the trigeminal nerves. In this case, if a direct action of cinnamaldehyde on oral microbiota toxin effects could be also conceivable in the Conclusions.

Response: In the introduction line 67 -75 has shown that periodontal pathogens can enter into the intestine and exerts its action. We have also previously reported that there was no evidence of periodontal pathogens in the blood stream. Our study mainly focuses on the intestinal barrier than the CNS. We agree to the reviewer suggestions and have emphasized in the conclusions that cinnamaldehyde a bioactive compound present in cinnamon can execute anti-microbial activity.

  1. Results: the first section of the results is confused (lines 97-113). Indeed, on the text results are described on separated Fig. 1a and Fig.1b, but Figure1 consists only of Fig 1A and Fig.1b is missing (please make also the letters uniform if uppercase or lowercase). Furthermore, in Figure 1 the histograms are too crowded and make it difficult to understand the data.

Response: We had submitted all the figures in a separate file also. As suggested, Figures 1A has been revised and separated into Fig 1A and Fig 1B, and all the letters has been uniformly assigned. Fig 1C – E, has been separated also based on two different concentrations of Pg-LPS with cinnamaldehyde pretreatment.

  1. In Figure 2, Authors should report the fluorescence intensity values read with vehicle, Pg-LPS and Pg in the caption of the figure.

Response: The authors have represented the fluorescence intensity values to percent control. In Fig 2A. It would be again an extra set of data to be presented in table form for all of the values. Hence, authors decided to keep the representative figure. However, the authors have given the intensity values for vehicle in the figure 2A legend (revised track-in changes manuscript line # 163-164).

  1. In section 2.3, lines 152-153, it is unclear what specific data of the results demonstrate that “pretreatment with CNM stimulated cell proliferation and increased protein levels and nuclear translocation of Nrf2”.

Response: It is clearly shown in Figures 2&3, that pretreatment with CNM enhances cell viability, suppresses keap 1 and activates Nrf2 protein expression which in turn upregulates Nqo1 the downstream of Nrf2 pathway. All this data authenticates the protective role of cinnamaldehyde in IEC 6 cells.

  1. Discussion: please, reference the statement of the line 243 “The GI tract is a key source of ROS, which are involved in many GI diseases.”, and the statement of the lines 280-281 “NOS catalyzes the synthesis of NO in vivo; under normal physiological states, NO synthesis mediated 280 by endothelial or gastric neuronal NOS serves a major role in the regulation of vascular or gastric motility.”.

Response: As suggested references for this statement has been added in the revised manuscript. Revised track-in changes manuscript line 283 (ref # 23), track-in-changes revised manuscript line 319-320 (ref # 35).

  1. Specify the meaning of the acronyms IBD (line 305) and NEC (line 306).

Response: Full forms for IBD (revised line 344) and NEC (revised line 345) expanded in the revised manuscript.

  1. Methods: line 330: Pg is of human or rat origin? I mean, if Pg is of human origin, is its effect on rat IEC-6 cells reliable? Authors should better specify why in their experimental design a human periodontal pathogen is tested on rat intestinal cells.

Response: Gingivalis bacteria used in this study was from human origin. P. gingivalis (W83) used in this study has shown to be highly virulent in experimental animal models. Moreover, the human intestine is more permeable of human intestine than in vitro models.  Hence, the experiments were conducted using rat IEC 6 cells. Our data is a preliminary study conducted to elucidate the mechanistic role of intestinal epithelial inflammation and barrier dysfunction. Further studies are in progress to see the effects of induction of periodontitis in-vivo and the protective role of cinnamaldehyde.

  1. Please, specify the meaning of the acronyms TSA (line 330), MOI (line 347) and FD4 (line 382) for the outsider readers.

Response: Acronyms elaborated for Tryptic Soy Agar (TSA) revised line 369, multiplicity of infection (MOI) revised line 386, and fluorescent-conjugated dye (FD4) revised line 422 in the revised manuscript.

  1. Section 4.4:  MTT solution was effectively incubated for 3 h at 5 mg/ml or this one was diluted 1:10?

Response: Thank you for bringing this our notice. There was typo, the final concentration is 0.5 mg/ml (line # 396). Reference for the protocol is also given in the revised manuscript (Ref # 50, revised line # 389).

  1. Conclusions: can it be concluded that the tested concentrations of cinnamaldehyde on IEC-6 cells are actually achievable in vivo in the intestine and, therefore, having the same effects also in the human intestine?

Response: Conclusion statement revised in the manuscript (revised line 463-464).

Reviewer 2 Report

Comments and Suggestions for Authors

The authors in the present study have highlighted the importance of Cinamaldehyde (CNM) against P. gingivalis infection causing oral periodontitis. They have concluded that CNM protected IEC-6 cells against Pg-induced oxidative damage and increased intestinal epithelial barrier permeability by activating the PI3K/Akt-mediated Nrf2/NQO-1 antioxidant signaling and suppressing inflammatory and apoptotic markers. The study is interesting and have a good potential for publication. I have following concens

1.     When mentioning intestinal epithelial barrier permeability please clearify that the study is in vitro and it has been done in a cell line that could be used for in vitro model intestinal epithelial barrier permeability. Please mention this in introduction and discussion. This is necessary for the readers proper understanding.

2.     Why data points are missing in some figure while its visible in some of the others. Please explain.

3.     Please mention that cell viability was decreased in 1ug of Pg LPS treatment in 72 hrs as evident in the data (Fig1a). Why is this concentration not used in subsequent experiments?

4.     Can’t find Fig. 1b (line 109)

5.     Please write the interpretation of the results as the bottom line in all the subsection of Results section

6.     P value indication with alphabets is confusing can the author use symbols or lines to indicate the comparison

7.     Fig 2 color coding (Red bars are used in two different groups Pg LPS 100ug/ml or Pg alone). The authors can use patterned bars to indicate the differences group depiction.

8.     Line 126-129, please mention what is the group being compared. Is it control or LPS or Pg alone.

9.     Fig 3B, in representative immunoblot there is an increase in Nrf2 protein level while actin remains the same upon Pg-LPS 10ug treatment without CNM. This is contradictory as Nrf2 protein should decrease upon LPS treatment. An opposite is seen in Pg-LPS 10 ug +CNM group.

10.  It seems pg 10ug LPS is not showing good response in terms of iNOS/ROS signaling. Pg 100 is more robust. The author can indicate this in text.  (Fig. 4A)

11.  The author can consider checking IL10 gene expression as an anti-inflammatory gene to strengthen there finding that Pg infection upregulates inflammatory pathway while CNM treatment rescue this.

12.  Data 5E-5G is missing.

13.  Line 198 please elaborate “TJs” as Tight junctions at the first introduction.

14.  Fig6A and Fig 6A. Representative blot don’t match the quantification. Please check.

15.  The author in the current study have shown that pretreatment with CNM has inhibited cellular apoptosis by modulating the Bcl-2/Bax/caspase-3 signaling pathway. Moreover, they have shown that iNOS is upregulated upon Pg infection. To strengthen their finding, they can cite paper where authors have shown that upon iNOS or nNOS upregulation there is an induction of proptosis and apoptosis in CML cell line.

Author Response

Reviewer # 2

Comments and Suggestions for Authors

The authors in the present study have highlighted the importance of Cinamaldehyde (CNM) against P. gingivalis infection causing oral periodontitis. They have concluded that CNM protected IEC-6 cells against Pg-induced oxidative damage and increased intestinal epithelial barrier permeability by activating the PI3K/Akt-mediated Nrf2/NQO-1 antioxidant signaling and suppressing inflammatory and apoptotic markers. The study is interesting and have a good potential for publication. I have following concerns

Response:  We are thankful for the valuable suggestions provided by this reviewer. We have carefully revised the manuscript and the figures as suggested.

            Comments:

  1. When mentioning intestinal epithelial barrier permeability please clearify that the study is in vitroand it has been done in a cell line that could be used for in vitro model intestinal epithelial barrier permeability. Please mention this in introduction and discussion. This is necessary for the readers proper understanding.

Response: The human intestine is more permeable of human intestine than in vitro models.  Hence, the experiments were conducted using rat IEC 6 cells. The statement has been written in the discussion section of the revised manuscript (line # 279-280).

  1. Why data points are missing in some figure while its visible in some of the others. Please explain.

Response: The authors think that Figure 1 & 2 are best represented as colored bar graph, where as Figure 3 onwards each individual data point were used to show how four different data points were separated with each other.

  1. Please mention that cell viability was decreased in 1ug of Pg LPS treatment in 72 hrs as evident in the data (Fig1a). Why is this concentration not used in subsequent experiments?

Response: The cell viability was significantly reduced at 72 hrs for 1ug of Pg-LPS however it was just 20 % reduction, so the authors selected other two concentrations for further experiments. Since, the other two concentrations showed protective role of cinnamaldehyde, it is obvious that at lower concentration would have the same results.

  1. Can’t find Fig. 1b (line 109)

Response: We had uploaded all of the manuscript figures separately also but some how the figures were not added in the template. In the revised manuscript we have revised all of the figures for clarity as well cross checked that they appear in the context.

  1. Please write the interpretation of the results as the bottom line in all the subsection of Results section

Response: As suggested, the authors have provided the interpretation at the end of each result.

  1. P value indication with alphabets is confusing can the author use symbols or lines to indicate the comparison

Response: Since the figures have been revised the symbols correlate to the comparisons.

  1. Fig 2 color coding (Red bars are used in two different groups Pg LPS 100ug/ml or Pg alone). The authors can use patterned bars to indicate the differences group depiction.

Response: The authors have revised the figures and separated them for better clarity.

  1. Line 126-129, please mention what is the group being compared. Is it control or LPS or Pg alone.

Response: As the figures have been revised, * were compared with control and # were compared to that with the infected cells either with LPS or Pg alone.

  1. Fig 3B, in representative immunoblot there is an increase in Nrf2 protein level while actin remains the same upon Pg-LPS 10ug treatment without CNM. This is contradictory as Nrf2 protein should decrease upon LPS treatment. An opposite is seen in Pg-LPS 10 ug +CNM group.

Response: The Pg-LPS did not have much effect on Nrf2 protein expression at 10ug without CNM however, the band density appears to be reduced but calculating band density with other representative blots we did not see much difference. Since this is also noted in our iNOS/ROS data as well.

  1. It seems pg 10ug LPS is not showing good response in terms of iNOS/ROS signaling. Pg 100 is more robust. The author can indicate this in text.  (Fig. 4A).

Response: Thank you for this suggestion, we have included this in the revised manuscript (line #199-200).

  1. The author can consider checking IL10 gene expression as an anti-inflammatory gene to strengthen there finding that Pg infection upregulates inflammatory pathway while CNM treatment rescue this.

Response: The authors will definitely look into this gene expression in our on going in vivo studies. Since these experiments were conducted

  1. Data 5E-5G is missing.

Response: We had uploaded all of the manuscript figures separately also but somehow, the figures were not added in the template. In the revised manuscript we have revised all of the figures for clarity as well cross checked that they appear in the context.

  1. Line 198 please elaborate “TJs” as Tight junctions at the first introduction.

Response: TJs had been already expanded in the introduction at line 72.

  1. Fig6A and Fig 6A. Representative blot don’t match the quantification. Please check.

Response: Thank you for bringing this to our notice. The authors have revisited the densitometric analysis and revised the figure accordingly.

Reviewer 3 Report

Comments and Suggestions for Authors

In this study, authors evaluated the protective role of cinnamaldehyde (CNM) against P. gingivalis and Pg-derived LPS induced intestinal epithelial barrier dysfunction in IEC-6 cells. Interesting, authors found that P. gingivalis or Pg-LPS increased ROS and MDA levels expressing oxidative stress damage. Moreover, Pg-LPS, as well as Pg alone, induced inflammatory cytokines activating TLR-4 signaling. Furthermore, infection reduced Nrf2 and NQO1 expression while iNOS expression increased. CNM treatment suppressed both Pg- and Pg-LPS-induced intestinal oxidative stress damage by reducing ROS, MDA, and NO production. Furthermore, CNM treatment upregulated the tight junction proteins expression. 

the manuscript is interesting and generally well written. The topic fits the aim of the journal. Figures must be improved. Tables are not present. 

Several points must be improved:

Lines 83-85: the multifaceted role of NRF2/KEAP1 signaling deserves to be highlighted since this pathway plays a key role also in cancer onset and progression (see PMID: 37175546).  This is an interesting point to add since it can further highlight the interesting results obtained by the authors.

Figure 1: This figure is quite confusing. I suggest to split it in two graphs. In particular, I suggest to put untreated, 25, 50, 100 µM of CNM in a dedicate graph and explain why authors used 100 µM of CNM.

4.10. Western blot analysis: I suggest to move the list of the antibodies used in a dedicate table

Figures remove "Gangula et al.," and figure number from all figures. Moreover, increase the images size in all figures

Line 392: Table 1 is missed

Abbreviations must be written in full length when mentioned for the first time

The manuscript must be formatted according to the journal style

Figures must be insert after they are mentioned for the first time

The N=3 for the calculation of p value could be insufficient for this type of experiments especially if experiments have been carried out in single

Author Response

Reviewer # 3

Comments and Suggestions for Authors

In this study, authors evaluated the protective role of cinnamaldehyde (CNM) against P. gingivalis and Pg-derived LPS induced intestinal epithelial barrier dysfunction in IEC-6 cells. Interesting, authors found that P. gingivalis or Pg-LPS increased ROS and MDA levels expressing oxidative stress damage. Moreover, Pg-LPS, as well as Pg alone, induced inflammatory cytokines activating TLR-4 signaling. Furthermore, infection reduced Nrf2 and NQO1 expression while iNOS expression increased. CNM treatment suppressed both Pg- and Pg-LPS-induced intestinal oxidative stress damage by reducing ROS, MDA, and NO production. Furthermore, CNM treatment upregulated the tight junction proteins expression. 

the manuscript is interesting and generally well written. The topic fits the aim of the journal. Figures must be improved. Tables are not present. 

Several points must be improved:

Response: The authors thank the reviewers for their valuable comments. We have addressed all the comments and suggestions in the revised manuscript.

Comments:

  1. Lines 83-85: the multifaceted role of NRF2/KEAP1 signaling deserves to be highlighted since this pathway plays a key role also in cancer onset and progression (see PMID: 37175546).  This is an interesting point to add since it can further highlight the interesting results obtained by the authors.

Response: Thank you for your suggestion. The authors have revised accordingly with the reference suggested (Ref # 11, line 83-84).

  1. Figure 1: This figure is quite confusing. I suggest to split it in two graphs. In particular, I suggest to put untreated, 25, 50, 100 µM of CNM in a dedicate graph and explain why authors used 100 µM of CNM.

Response: As suggested, the figures has been revised with Fig 1A, Fig 1B and Fig 1C-E.

  1. Section 4.10. Western blot analysis: I suggest to move the list of the antibodies used in a dedicate table

Response: As suggested, the authors have now revised the manuscript by having a dedicated table assigned to the list of antibodies (Table 2).

  1. Figures:  remove "Gangula et al.," and figure number from all figures. Moreover, increase the images size in all figures

Response: Gangula et al., at the end of all the figures as been removed in the revised manuscript.

  1. Line 392: Table 1 is missed

Response: Thank you for bringing this to our notice. We now have provided the missing table 1.

  1. Abbreviations must be written in full length when mentioned for the first time

Response: The authors have cross checked for abbreviations and expansion in the revised manuscript.

  1. The manuscript must be formatted according to the journal style

Response: The manuscript as well as the references arranged as per the journal guidelines.

  1. Figures must be insert after they are mentioned for the first time

Response: Figures are arranged accordingly within the text of the revised manuscript.

  1. The N=3 for the calculation of p value could be insufficient for this type of experiments especially if experiments have been carried out in single

Response: N=3 is three independent experiments and, in each experiment, we had triplicates values for all of the results.

Round 2

Reviewer 1 Report

Comments and Suggestions for Authors

I agree with your careful review of the manuscript.

Reviewer 3 Report

Comments and Suggestions for Authors

the manuscript has been significantly improved and can be accepted in the present form